# Explain Any Concept: Segment Anything Meets Concept-Based Explanation

**Ao Sun, Pingchuan Ma, Yuanyuan Yuan, and Shuai Wang**
The Hong Kong University of Science and Technology
`{asunac, pmaab, yyuanaq, shuaiw}@cse.ust.hk`

## Abstract

EXplainable AI (XAI) is an essential topic to improve human understanding of deep neural networks (DNNs) given their black-box internals. For computer vision tasks, mainstream pixel-based XAI methods explain DNN decisions by identifying important pixels, and emerging concept-based XAI explore forming explanations with concepts (e.g., a head in an image). However, pixels are generally hard to interpret and sensitive to the imprecision of XAI methods, whereas "concepts" in prior works require human annotation or are limited to pre-defined concept sets. On the other hand, driven by large-scale pre-training, Segment Anything Model (SAM) has been demonstrated as a powerful and promotable framework for performing precise and comprehensive instance segmentation, enabling automatic preparation of concept sets from a given image. This paper for the first time explores using SAM to augment concept-based XAI. We offer an effective and flexible concept-based explanation method, namely Explain Any Concept (EAC), which explains DNN decisions with any concept. While SAM is highly effective and offers an "out-of-the-box" instance segmentation, it is costly when being integrated into de facto XAI pipelines. We thus propose a lightweight per-input equivalent (PIE) scheme, enabling efficient explanation with a surrogate model. Our evaluation over two popular datasets (ImageNet and COCO) illustrate the highly encouraging performance of EAC over commonly-used XAI methods.

## 1  Introduction

In recent years, Deep Neural Networks (DNNs) have exhibited exceptional performance in a variety of computer vision (CV) tasks such as image classification [1], object detection [2], and semantic segmentation [3]. However, due to the "black-box" nature of these complex models, their use in security-sensitive applications where interpretability is critical is still limited. As a result, there is a growing demand for increased transparency and comprehensibility in the decision-making processes of DNNs. To address this issue, Explainable AI (XAI) [4] has emerged with the purpose of providing explanations for DNNs' predictions.

For CV tasks, conventional XAI works primarily focus on proposing and enhancing pixel-level interpretation, which explains the model prediction by identifying important pixels. Despite the strides made in XAI, these techniques often involve trade-offs between three key desiderata among a bunch of criteria: *faithfulness*, *understandability*, and *efficiency* [5, 6, 7, 8]. Faithfulness ensures that the generated explanation aligns with the internal decision-making process of the DNN, understandability ensures that the explanations are human-comprehensible, and efficiency guarantees that the explanations are generated with a reasonable computational cost.

From the perspective of explanation forms, existing methods often provide pixel or superpixel-level explanations [9, 10, 11, 12, 13], which are constantly hard to interpret (i.e., low understandability) and sensitive to the potential imprecision of XAI techniques (low faithfulness). Some recent works

aim to explain DNN predictions with concepts (e.g., a head rather than pixels in the head) in images. However, they either require human annotation or are limited specific concept discovery methods.

From the XAI methodology perspective, Shapley value [14]-based explanation has become the mainstream, given its well-established guarantee in theory. However, the inherent complexity of Shapley value calculations and the target model makes it highly costly and time-consuming (*low efficiency*). To reduce the high overhead, existing methods rely on Monte Carlo sampling [15], model-specific approximations [16, 17], (smaller) surrogate models [18, 19], or a combination of the above to approximate the Shapley value. With the growth of model size and complexity, surrogate models are widely employed, even though it may suffer from low faithfulness owing to the discrepancy between the surrogate model and the target model under consideration.

**Our Solution.** Driven by large-scale pre-training, Segment Anything Model (SAM) [20] has been demonstrated as a powerful and promotable framework for performing precise and comprehensive instance segmentation, enabling automatic extraction of a concept set from an given image. Hence, this paper for the first time explores using SAM as a concept discovery method to augment concept-based XAI. We advocate to line up SAM and XAI, such that SAM's instance segmentation delivers high accurate and human-understandable concept set from arbitrary images, which in turn facilitates the XAI task with high faithfulness and understandability.

Nevertheless, while SAM is effective and offers an "out-of-the-box" solution, computing the Shapley value is still expensive. Thus, to achieve high efficiency (our third desiderata), besides standard Monte Carlo sampling method to reduce overheads, we propose a lightweight *per-input equivalent (PIE) scheme* to approximate the target model with a low-cost surrogate model. Our PIE scheme allows the surrogate model to share some carefully-chosen parameters with the target model, which effectively close the discrepancy between the two models.

Our evaluation over two popular datasets (ImageNet and COCO) illustrate the highly encouraging and superior accuracy of EAC over popular pixel-level and superpixel-based XAI methods. Moreover, we also demonstrate the high interpertertability of EAC with a carefully-designed user study. As confirmed by human experts, EAC offers high interpretability, largely outperforming de facto XAI methods. We also justify the effectiveness of our technical pipeline with ablation studies, and discuss potential extension and future work. In summary, this paper makes the following contributions:

- We for the first time advocate the usage of SAM as a concept discovery method to facilitate concept-based XAI with high faithfulness and understandability.

- We propose a general and flexible concept-based explanation pipeline, namely Explain Any Concept (EAC), which can explain the model prediction with any concept. We introduce a set of design considerations and optimizations to make EAC practically efficient while maintaining high faithfulness and understandability. In particular, to reduce the generally expensive computational burden of Shapley Value, we propose the a novel scheme named Per-Input Equivalence (PIE).

- We conduct extensive experiments and human studies to demonstrate the effectiveness of EAC on diverse settings. We also illustrate the generalizability of EAC and its security benefits.

**Open Source.** We publicly release and maintain EAC under the following github page: https://github.com/Jerry00917/samshap.

## 2 Background and Related Works

EAC belongs to local XAI which offers model-agnostic explanations for DNN decisions over each image input. Below, we introduce prior works from both XAI methodology and input data perspective.

**XAI Methodology.** From the methodology perspective, XAI can be classified into two categories: backpropagation-based and perturbation-based. The former case, also known as gradient-based, leverages the backward pass of a neural network to assess the influence and relevance of an input feature on the model decision. Representative works include saliency maps [9], Gradient class activation mapping (Grad-CAM) [11], Salient Relevance (SR) maps [21], Attributes Maps (AM) [22], DeepLIFT [23], and GradSHAP [12]. For the latter case, it primarily perturbs the input data into variants, and then measures the change in model output to generate explanations. Representative

works include LIME [13], SHAP [18], and DeepSHAP [18]. Given that EAC employs Shapley value, we detail it in the following.

*Shapley Value for XAI.* The Shapley value [14], initially introduced in the cooperative game theory, quantifies each player's contribution to the total payoff of a coalition. This concept has been adapted to machine learning to measure each feature's contribution to a model's prediction. The Shapley value for player $i$ is determined by their average marginal contribution across all possible coalitions, taking into account all potential player orderings. Formally, it can be expressed as: $\phi_i(v) = \frac{1}{N} \sum_{k=1}^{N} \frac{1}{\binom{N-1}{k-1}} \sum_{S \in S_k(i)} (u(S \cup \{i\}) - u(S))$ where $N$ represents the set of all players, $S_k(i)$ is the collection of all coalitions of size $k$ that include player $i$, $u$ is the utility function, and $S$ is a coalition of players. In short, the Shapley value enumerates all possible coalitions and calculates the marginal contribution of the player $i$ in each coalition. The Shapley value possesses several desirable attributes, including efficiency, symmetry, and additivity, and uniquely satisfies the property of locality. According to this property, a player's contribution depends solely on the members involved in the respective coalition.

In the context of XAI, $N$ often represents the size of feature space, $S$ is a subset of features, and $u$ is the model's prediction. For instance, consider the pixel-level Shapley value for image classification. In this case, $N$ is the number of pixels in the image, $S$ is a subset of pixels, and $u(S)$ is the probability of the image belonging to a particular class when all pixels excluding $S$ are masked. Likewise, in superpixel-based and concept-based methods, $N$ is the number of superpixels or concepts, $S$ is a subset of super-pixels or concepts, and $u(S)$ is the prediction of the model when the remaining superpixels or concepts are masked.

**Different Forms of Explanations.** Given an input image, existing XAI techniques may explain the DNN decision at the level of individual pixels, superpixels, or concepts.

*Pixel-Based XAI* explains DNN decision at the level of individual image pixels. Prior works often use saliency maps [9, 10], which highlight the most important pixels for a given decision. Activation maximization [24] modifies an input image to maximize the activation of a particular neuron or layer in a DNN to explain its decision. Attention [25, 26] is also employed in XAI, offering relatively low-cost explanation in comparison to standard gradient-based methods. However, these techniques have limitations, such as being highly dependent on the model architecture and not providing a complete understanding of the decision-making process.

*Superpixel-Based XAI* explains DNN decision at the level of super-pixels, which are often perceptual groups of pixels. SLIC [27] is a common superpixel algorithm that clusters pixels together based on their spatial proximity and color similarity. SLIC forms the basis of various mainstream XAI methods, such as Class Activation Mapping (CAM) [11] and Grad-CAM [12]. LIME offers a model agnostic approach to XAI that explains model decisions by training a local explainable model over superpixels. RISE [28] randomly masks parts of the input image and observes the model prediction change. Superpixels that have the most significant impact on the output are then used to create an importance map for explanation. Superpixel-based methods group pixels together to simplify the image, but this may result in a loss of resolution and detail, particularly when attempting to identify small or intricate details in an image. The accuracy of superpixel-based methods is highly dependent on the quality of the image being analyzed, and more importantly, objects that are recognized in the image. Ill-conducted superpixel segmentation can notably undermine the accuracy of the explanation.

*Concept-Based XAI* leverages concept activation vectors (CAVs) [29] extracted from neuron activations to distinguish between images containing user-specified concepts and random images. It offers a quantitative way of depicting how a concept influences the DNN decision. Furthermore, the CAV method is extended in recent works with more flexibility and utility (e.g., with causal analysis or Shapley values) [30, 31, 32]. However, these techniques generally require human annotation or are limited to a pre-defined set of concepts, and this paper advocates a general and flexible concept-based XAI technique.

# 3  Method

## 3.1  Desiderata of XAI Techniques

XAI aims to address the opaque nature of DNNs and provide explanations for their predictions. Aligned with prior research [5, 6, 7], we outline three essential desiderata for XAI techniques applied in computer vision tasks.

**Faithfulness.** The explanation should be highly faithful with respect to the internal mechanism of the target DNN. Let $E$ be the explanation and $f$ be the target DNN model under analysis, the faithfulness can be defined as the correlation between $E$ and the actual decision-making process of $f$. A higher correlation indicates a more effective explanation. To date, a consensus is yet to be reached on how to quantitatively measure faithfulness [28, 33, 34, 35]. In this paper, we employ insertion/deletion experiments [28], a frequently-used approach to assessing faithfulness.

**Understandability.** Although faithfulness is a necessary element, it alone does not suffice for a desirable XAI process. It is equally crucial that the explanation delivered by the XAI process is comprehensible to humans. In other words, the XAI process must provide a lucid and human-understandable explanation that allows users to easily grasp the reasoning behind the model's decision-making process.

**Efficiency.** The explanation generation process should have relatively low computational cost so as not to cause unacceptable delays. This ensures that the XAI technique can be practically implemented in real-world applications without compromising system performance.

**Trade-off.** Note that the three desiderata are frequently in conflict with each other. For example, a highly faithful explanation, strictly reflecting the internal mechanism of the target DNN, may be too obscure to understand. Namely, a too large superpixel size can yield inaccurate yet more "complete" (and thus more readable) outputs, and vice versa for a too small superpixel size. In fact, a proper superpixel size is often hard to decide. Moreover, faithfulness may also impose a significant challenge to efficiency. In fact, generating a explanation that faithfully explains the target model is usually costly, especially for models with popular large backbones. Hence, many existing XAI techniques trade faithfulness for efficiency by using surrogate models to approximate the target model. In sum, this paper advocates to strike a balance between the three desiderata to obtain an effective XAI technique, as will be illustrated in our technical design (Sec. 3) and evaluation (Sec. 4).

## 3.2  EAC Design

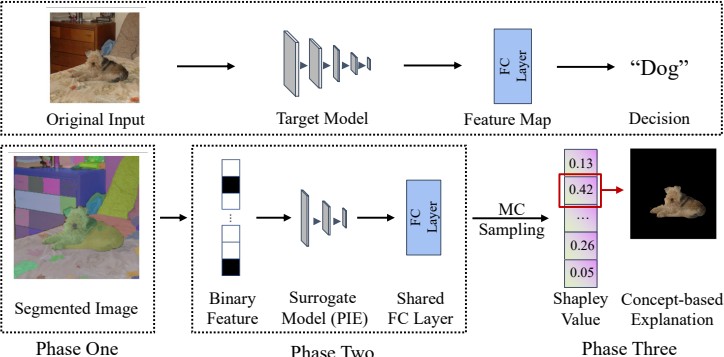

Figure 1: The technical pipeline of EAC in a three-phase form.

**Technical Pipeline.** As depicted in Fig. 1, EAC features a three-phase pipeline to explain a DNN's prediction for an input image. In the first phase, we employ the de facto instance segmentation model, SAM, to partition an input image into a set of visual concepts. In the second phase, we train a per-input equivalent (PIE) surrogate model to approximate the behavior of the target DNN. In the third phase, we use the surrogate model to efficiently explain the model prediction with the concepts obtained in the first phase. In sum, EAC lines up SAM with XAI, such that the de facto instance segmentation model forms the basis of XAI *faithfulness* and *understandability*, whereas our

novel PIE scheme offers high *efficiency* while maintaining high *faithfulness*. Below, we present the technical pipeline of EAC in detail.

**Phase One: Concept Discovery.** Concepts are defined as prototypes that are understandable for humans [36, 37]. Traditionally, methods such as ACE [31] and CONE-SHAP [5] leverage superpixel method on the validation set. Then, these superpixels are clustered into a set of concepts using its feature vectors. However, this does not necessarily render the concepts that are understandable for humans. In this regard, we advocate that the concepts should be semantically meaningful and human-understandable. To this end, we employ the de facto instance segmentation model, SAM, to obtain the concepts. Given an input image $x$, SAM outputs a set of instances in the image and these instances constitute the concept for $x$. We present a sample case in "Phase One" of Fig. 1. Here, we denote the set of concepts as $\mathcal{C} = \{c_1, c_2, \cdots, c_n\}$, where $n$ is the number of concepts.

**Phase Two: Per-Input Equivalence (PIE).** Aligned with recent research in XAI, we use the Shapley value [14] to identify key concepts that contribute to the target model's prediction; the design detail is presented below. However, despite the general effectiveness of Shapley value, it is highly costly to compute due to the exponential complexity. Overall, the Shapley value needs to enumerate all possible coalitions and calculates the marginal contribution for each concept. While the exponential complexity is usually avoided by Monte Carlo sampling, it is still costly due to the inherent complexity of the target model (e.g., models with large backbones).

To alleviate this hurdle, we propose the scheme of *Per-Input Equivalence* (PIE) to reduce the complexity of the target model. Intuitively, we would expect to substitute the target model with a surrogate model $f'$ that is computationally efficient while maintaining the same functionality of the target model. However, it is challenging to obtain a simple model $f'$ that is fully equivalent to the target DNN model $f$. Since each time we only explain the target model for one certain input, we can employ a surrogate model $f'$ that is only equivalent to $f$ over the given certain input (i.e., PIE). Therefore, we have $f'(\boldsymbol{b}) := f_{\text{fc}}(h(\boldsymbol{b}))$, where $\boldsymbol{b}$ is the one-hot encoding of the concepts in $\mathcal{C}$, $h$ mimics the feature extractor of $f$, and $f_{\text{fc}}$ is the fully-connected (FC) layer in $f$.

Formally, we have $h : \{0,1\}^{|\mathcal{C}|} \to \mathbb{R}^m$ where $m$ is the size of features in the last layer of $f$. Considering the "Phase Two" in Fig. 1, where to constitute the PIE scheme, $f'$ takes the one-hot encoding of the concepts as input, leverages $h$ to extract the feature vector with the same semantics as $f$, and predicts the expected output of $f$ using the same FC layer (referred to as "Shared FC Layer" in Fig. 1) of $f$. The training data for $f'$ is obtained by sampling the concepts in $\mathcal{C}$ and the corresponding probability distribution of $f$ by masking the concepts in $\mathcal{C}$. Then, we can train $f'$ by plugging the FC layer of $f$ into $f'$ as freezed parameters and optimize $h$ with the cross-entropy loss.

Hence, we only use the target model $f$ to obtain a few training samples for our surrogate model $f'$ and then use $f'$ to explain $f$. Given that $f'$ is much smaller than $f$, the PIE scheme can significantly reduce the cost of computing the Shapley value, enabling efficient Shapley value calculations for $f$.

**Phase Three: Concept-based Explanation.** Given a set of visual concepts $\mathcal{C} = \{c_1, c_2, \cdots, c_n\}$ identified in the first phase, we aim to explain the model prediction for a given input image $x$. The explanation can be expressed as a subset of $\mathcal{C}$, i.e., $E \subseteq \mathcal{C}$. Recall the definition of Shapley value in Sec. 2, we consider each concept as a player and the model prediction on the original class of $x$ as the utility function. Then, we can define the marginal contribution of a concept $c_i$ as the difference between the model prediction on $S \cup \{c_i\}$ and $S$ where $S \subseteq \mathcal{C} \setminus \{c_i\}$. That is, the marginal contribution of $c_i$ is defined as

$$\Delta_{c_i}(S) = u(S \cup \{c_i\}) - u(S) \tag{1}$$

Here, $u(S) := f(\text{mask}(x, \mathcal{C} \setminus S))$ is the prediction of the target model $f$ on the image with only concepts in $S$ (remaining concepts are masked). With the aforementioned PIE scheme, we can use the surrogate model $f'$ to approximate $f$ (i.e., $\hat{u}(S) := f'(\text{mask}(x, \mathcal{C} \setminus S))$). Then, the Shapley value of $c_i$ is defined as

$$\phi_{c_i}(x) = \frac{1}{n} \sum_{k=1}^{n} \frac{1}{\binom{n-1}{k-1}} \sum_{S \in S_k(i)} \Delta_{c_i}(S) \tag{2}$$

where $S_k(i)$ is the collection of all coalitions of size $k$ that does not contain $c_i$. Since the size of all coalitions is prohibitively large, we approximate the Shapley value using Monte Carlo (referred to as "MC Sampling" in Fig. 1) sampling. In particular, we sample $K$ coalitions for each concept

and approximate the Shapley value as $\hat{\phi}_{c_i}(x) = \frac{1}{K} \sum_{k=1}^{K} \Delta_{c_i}(S_k)$, where $S_k$ is the $k$-th sampled coalition. The optimal explanation is defined as the subset of concepts that maximizes the Shapley value, i.e.,

$$E = \arg_{E \subset \mathcal{C}} \max \hat{\phi}_E(x) \tag{3}$$

where $\hat{\phi}_E(x) = \sum_{c_i \in E} \hat{\phi}_{c_i}(x)$ is the Shapley value of $E$ for $x$. Finally, we mask the concepts in $\mathcal{C} \setminus E$ and provide the masked image as the visual explanation to the user.

**Comparison with Existing Surrogate Models.** We are aware of existing methods such as LIME [13] that also use a low cost surrogate model to mimic the target model. We highlight the key differences between our method and LIME. First, LIME uses a linear model as the surrogate model, which is not expressive enough to approximate the target model. Second, LIME learns the relation between the input and output of the target model. In contrast, our method effectively reuses the FC layer of the target model $f$ to form the surrogate model $f'$. This shall generally enable more accurate approximation of the target model as the FC layer is retained.

# 4 Evaluation

In this section, we evaluate EAC from three aspects: (1) the faithfulness of EAC in explaining the model prediction, (2) the understandability of EAC from the human perspective, and (3) the effectiveness of PIE scheme compared with the standard Monte Carlo sampling and surrogate model-based methods.

## 4.1 EAC Faithfulness

**Setup.** We evaluate EAC on two popular datasets, ImageNet [38] and COCO [39] and use the standard training/validation split for both datasets. We use the ResNet-50 [1] pre-trained on ImageNet/COCO as the target DNN model for EAC. We present discussion on generalizability of EAC in Sec. 5 and corresponding empirical evaluations in Supplementary Materials. We clarify that EAC does not involve many hyper-parameters. The only hyper-parameter considered in EAC is when fitting the PIE Scheme, i.e. a simple linear neural network learning scheme and the Monte Carlo (MC) sampling: we set $lr = 0.008$ and the number of MC sampling as 50000 throughout all experiments.

We use the *insertion* and *deletion* schemes to form our evaluation metrics; these two schemes are commonly used in the literature for evaluation of XAI techniques [28]. To clarify, these metrics involve generating predictions by gradually inserting and deleting concept features from the most important to the least important ones, and then measuring the Area Under the Curve (AUC) of prediction probabilities. Particularly, for insertion, it starts with a fully masked image and gradually reveals the concepts, while for deletion, it starts with a fully unmasked image and gradually masks the concepts. Intuitively, the AUC reflects the impact of the inserted/deleted concepts on the model prediction. For insertion, higher AUC indicates better faithfulness, while for deletion, lower AUC indicates better faithfulness. For both settings, we report the average results and standard deviation of three random runs.

**Baselines.** We compare EAC with nine baseline methods: (1) DeepLIFT [23] and (2) GradSHAP [12] are two representative backpropagation-based methods, which have been noted in Sec. 2. (3) Int-Grad [40], a gradient-based method, yields explanation by computing the path integral of all the gradients in the straight line between an input and the corresponding reference input; (4) Kernel-SHAP [18] approximates Shapley values by solving a linear regression problem; (5) FeatAbl (Feature Ablation) is a frequently-used perturbation based approach for feature attribution [41]; (6) CRAFT [42] is an attribution-based method allowing heatmaps to identify the most influential regions of an image. (7) LIME [13], a popular perturbation-based method introduced in Sec. 2, uses a surrogate model to approximate the target model. To apply these methods, we first employ superpixel [27] to generate concept patches from the input image, and then calculate their concept importance. The details of the superpixel hyper-param setting and the corresponding experiment are provided in the Supplementary Material.

Besides, we also compare EAC with three variants of DeepLIFT, GradSHAP, and IntGrad, which leverages morphological operations [43] (e.g., image erosion and dilation [44]) as a postprocessing step to cluster pixels into concepts. We denote these variants as DeepLIFT*, GradSHAP*, and

IntGrad*, respectively. Overall, we compare EAC with six superpixel-based XAI methods and three postprocessing-based concept XAI.

Table 1: Comparison with baseline methods across four settings. For each setting, the first and the second row report the mean and std. dev. of the results of three runs, respectively. ↑ and ↓ indicate higher and lower is better, respectively.

| | EAC | DeepLIFT | GradSHAP | IntGrad | KernelSHAP | FeatAbl | LIME | CRAFT | DeepLIFT* | GradSHAP* | IntGrad* |
|---|---|---|---|---|---|---|---|---|---|---|---|
| ImageNet/Insertion ↑ | **83.400** | 75.235 | 64.658 | 68.772 | 64.544 | 70.187 | 76.638 | 60.40 | 14.707 | 14.794 | 15.120 |
| | 0.023 | 0.000 | 0.035 | 0.000 | 0.002 | 0.000 | 0.027 | 0.000 | 0.000 | 0.067 | 0.000 |
| CoCo/Insertion ↑ | **83.404** | 78.199 | 61.109 | 65.037 | 54.570 | 72.260 | 79.028 | 51.49 | 8.580 | 21.643 | 19.755 |
| | 0.012 | 0.000 | 0.212 | 0.000 | 0.004 | 0.000 | 0.061 | 0.000 | 0.000 | 0.094 | 0.000 |
| ImageNet/Deletion ↓ | **23.799** | 25.262 | 40.996 | 36.214 | 26.583 | 37.332 | 25.307 | 54.66 | 40.620 | 44.830 | 46.015 |
| | 0.005 | 0.000 | 0.061 | 0.000 | 0.034 | 0.000 | 0.064 | 0.000 | 0.000 | 0.246 | 0.000 |
| CoCo/Deletion ↓ | **16.640** | 17.026 | 34.038 | 30.074 | 20.054 | 26.535 | 17.337 | 44.93 | 49.697 | 35.302 | 38.148 |
| | 0.041 | 0.000 | 0.144 | 0.000 | 0.040 | 0.000 | 0.049 | 0.000 | 0.000 | 0.173 | 0.000 |

We report the effectiveness of EAC and compare it with the baseline methods in Table 1. In particular, we observe that EAC consistently outperforms the baseline methods across all settings. For example, in the ImageNet dataset, EAC achieves 83.400% AUC for insertion, which is 8.165% higher than the second-best method DeepLIFT. Similarly, in the COCO dataset, EAC achieves 83.404% AUC for insertion, which is 5.205% higher than the second-best method DeepLIFT. We observe similarly promising results for the deletion evaluations. We also observe that the standard deviation of EAC is comparable to these baseline methods, which indicates that EAC is as stable as the majority of the baseline methods. Moreover, it is seen that the variants of DeepLIFT, GradSHAP, and IntGrad perform much worse than the original methods, which indicates the incapability of those standard morphological operations in clustering pixels, and the superiority of SAM in concept discovery.

## 4.2 EAC Understandability

To further explore the understandability of outputs from different XAI methods, we conduct a human evaluation to assess whether EAC can generate more human-understandable explanations than the baseline methods. To do so, we randomly select 100 images from the ImageNet and COCO datasets, respectively. We then generate eight explanations for each image using EAC and the seven baseline methods (for those three morphological operation-based variants in Table 1, we pick the best one in this evaluation). Then, for each image, we shuffle the explanations generated by EAC and the baseline methods, and ask our human participants (see below) to pick an explanation that they think is most understandable.

At this step, we recruit six participants to evaluate the explanations. We clarify that all participants are graduate students in relevant fields. We spent about 15 minutes to train the participants to understand our task. Before launching our study, we also provide a few sample cases to check whether the participants understand the task. To reduce the workload, for each image and its associated eight explanations, we randomly select three participants for the evaluation. Thus, each participant needs to evaluate 100 explanations. We report that each participant spends about 45 minutes to finish the evaluation.

Table 2: Human evaluation results.

| | EAC | DeepLIFT | GradSHAP | IntGrad | KernelSHAP | FeatAbl | Lime | GradSHAP* | Discord |
|---|---|---|---|---|---|---|---|---|---|
| ImageNet | 70 | 6 | 3 | 4 | 5 | 1 | 2 | 2 | 7 |
| COCO | 67 | 7 | 3 | 5 | 3 | 5 | 1 | 0 | 9 |

We report the results in Table 2. Among 200 images in total, participants reach a consensus (i.e., at least two out of three participants favors the same explanation) on 184 images (92.0%) with 93 images from ImageNet and 91 images from COCO. Among these 184 images, EAC is favored on 137 images (74.5%) while the second best baseline method is only favored on seven images (7.7%). From Table 2, it is seen that among the baseline methods, no one is significantly better than the others.

Overall, we conclude that EAC is significantly better than the baseline methods in terms of understandability. This is an encouraging finding that is in line with our study in Table 1. Below, we present several cases to demonstrate the EAC's superiority in understandability.

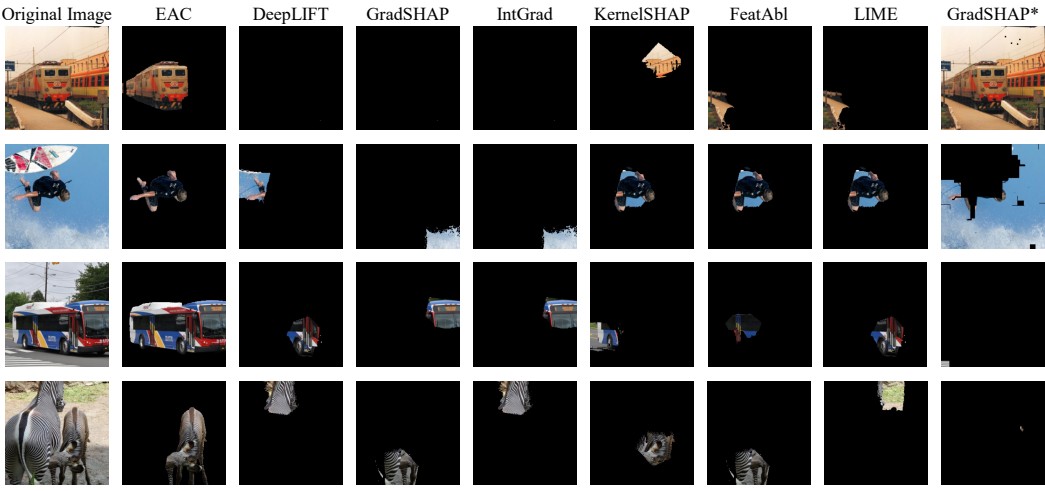

| Original Image | EAC | DeepLIFT | GradSHAP | IntGrad | KernelSHAP | FeatAbl | LIME | GradSHAP* |

Figure 2: Sample explanations generated by EAC and the baseline methods.

**Case Study.** We present several cases in Fig. 2 to demonstrate the effectiveness of EAC in explaining the model prediction. Besides these four cases, we observe that EAC consistently demonstrates its superiority in explaining the model prediction across other test cases. Overall, it is clear that EAC generates more "well-formed," concept-level explanations that are human understandable across all four cases in Fig. 2. For example, in the first case, EAC correctly highlights the "train" as the concept-level explanation, whereas the baseline methods yield some negligible pixels (the first three baselines), a fragment of the image (the 4th, 5th, and 6th baselines), or the entire image (the 7th baseline). This clearly illustrates the superiority of EAC in terms of the faithfulness and understandability.

### 4.3 Ablation Study of the PIE Scheme

In this experiment, we explore the effectiveness of our proposed PIE scheme. We compre the PIE scheme with three baselines that compute the Shapley value using 1) the original model under explanation, 2) a standard linear surrogate model and, 3) the same model in our PIE scheme while parameter sharing is disabled. In all settings, we use the same Monte Carlo sampling method to approximate the Shapley value while using different schemes to represent the target DNN model.

First, for the setting of directly using the original model, we observe a significantly longer processing time than the others. Indeed, we report that it takes more than 24 hours to process one image when using the same Monte Carlo sampling method. When we slightly reduce the number of samples, the processing time is still much longer than the other three methods (two baselines and EAC). Similarly, we find that the AUC is also much lower than that of the other three methods. As a result, we deem that the original model is impractical for computing the Shapley value and a surrogate model is necessary. Accordingly, we omit reporting the results of the original model and mark its results as "N/A" in Table 3.

Table 3: Ablation study of the PIE scheme. For each setting, we report the the mean of the results of ten runs. Aligned with Table 1, ↑ and ↓ indicate higher and lower is better, respectively.

| | Model | AUC | Processing Time (sec.) | | | Model | AUC | Processing Time (sec.) |
|---|---|---|---|---|---|---|---|---|
| ImageNet/Insertion ↑ | PIE | 81.78 | 245 | | ImageNet/Deletion ↓ | PIE | 12.47 | 244 |
| | Original Model | N/A | N/A | | | Original Model | N/A | N/A |
| | PIE w/o PS | 50.40 | 288 | | | PIE w/o PS | 32.87 | 289 |
| | Linear Model | 78.11 | 36 | | | Linear Model | 14.08 | 31 |
| COCO/Insertion ↑ | PIE | 87.08 | 252 | | COCO/Deletion ↓ | PIE | 13.71 | 203 |
| | Original Model | N/A | N/A | | | Original Model | N/A | N/A |
| | PIE w/o PS | 42.86 | 250 | | | PIE w/o PS | 38.36 | 222 |
| | Linear Model | 74.86 | 67 | | | Linear Model | 14.36 | 131 |

We report the results of our PIE scheme and the baselines in Table 3. Overall, we interpret the evaluation results as highly encouraging: the PIE scheme is notably better than all three baselines. In particular, the ablated PIE scheme without parameter sharing (the "PIE w/o PS" rows in Table 3) is

significantly worse than the PIE scheme in terms of both AUC and the processing time. This indicates that parameter sharing is effective in reducing the processing time while preserving high accuracy. Moreover, when comparing with the linear surrogate model (the "Linear Model" rows in Table 3), the PIE scheme is consistently better in terms of AUC. This indicates that the PIE scheme is more accurate, because the linear surrogate model over-simplifies the target model. Overall, we interpret that this ablation study illustrates the necessity of our PIE scheme, which empowers high faithfulness and efficiency to our technical pipeline.

## 5   Discussion

In this section, we first analyze the generalizability of EAC from the following three aspects. We then present statements on the security impact of EAC. Evaluations noted in this section are presented in the Supplementary Material.

**Different Target Models.** In this paper, we primarily use ResNet-50 as the target DNN model to evaluate EAC. Nevertheless, it is evident that the technical pipeline of EAC is independent of particular target DNN models. Further evaluation of EAC's performance using different target DNN models is presented in the Supplementary Material. In short, our findings demonstrate the persistent superiority of EAC over baselines when explaining different target DNN models.

**Different Visual Domains.** The concept extraction process for EAC is executed using the SAM framework, which means that it functions optimally on identical visual domains to SAM. It is advocated that SAM can be effective on a wide range of visual domains, including medical images, simulation images, and painting images [20]. However, our preliminary study over some other visual domains illustrate potentially suboptimal performance of SAM. Overall, it is clear that the accuracy of the concept set identified by SAM significantly impacts the performance of EAC, and ill-identified concepts may lead to suboptimal results of EAC. We remain exploring further applications of EAC on other visual domains for future research. Also, with the recent development of knowledge-specific area SAMs, such as medical image [45], remote sensing [46], and UVA [47], we believe that EAC has the potential to improve DNN decisions in other targeted areas in the future. Since this is the first attempt to deliver humanly understandable and computationally feasible concept extractors to the field XAI, we believe our work can shed light on their explainability.

**Different Visual Tasks.** This paper illustrates the effectiveness of EAC on interpreting image classification, a common and core task widely studied in previous XAI research. However, EAC is not limited to image classification. In fact, the technical pipeline of EAC can be applied and extended to other common visual tasks that can be explained using Shapley value-based explanations, such as object detection. We leave exploring EAC's potential applications on other visual tasks as future work.

**Security Impact.** Analyzing and detecting backdoors in DNNs is an emerging downstream application of XAI techniques. While the main paper primarily focuses on the core functionality of EAC, we also consider the security impact of EAC. In particular, we present evaluations of using EAC to analyze and detect backdoors in DNNs in the Supplementary Material. We report promising results and present discussions on potential future improvement accordingly.

## 6   Conclusion and Impact Statement

In this paper, we propose EAC, a novel method for explaining the decision of a DNN with any concept. EAC is based on SAM, a powerful and promotable framework for performing precise and comprehensive instance segmentation over a given image. We propose a highly efficient neural network pipeline to integrate the SAM and shapley value techniques. We conduct extensive experiments to demonstrate the effectiveness and interpretability of EAC.

**Broader Impact.** EAC is a general framework for explaining the prediction of a DNN with any concept. It can be used in many applications, such as medical image analysis, autonomous driving, and robotics. For example, EAC can be used to explain the prediction of a DNN for a medical image with any concept, such as a tumor. This can help doctors to understand the prediction of the DNN and improve the trustworthiness of the DNN. Overall, we believe that EAC can be used to improve

the trustworthiness of DNNs in many applications, and we publicly release and maintain EAC to facilitate its adoption in the real world.

However, in terms of **Negative Societal Impacts**, EAC may also be used to explain the prediction of a DNN for a medical image with a concept that is not related to the medical image, such as a cat, a train, or some subtle errors. This may mislead doctors and cause serious consequences. Therefore, it may cause harm when using EAC in real-world sensitive domains without sufficient safety checks on its outputs. We will continue to improve EAC to reduce its negative societal impacts.

## Acknowledgement

We thank the anonymous reviewers for their insightful comments. This research is supported in part by the HKUST 30 for 30 research initiative scheme under the the contract Z1283.

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

## Different Target Models

Table 4: ViT-b16, 50 runs

|  | EAC | DeepLIFT | GradSHAP | IntGrad | KernelSHAP | FeatAbl | LIME |
|---|---|---|---|---|---|---|---|
| ImageNet/Insertion ↑ | **89.594** | 54.455 | 68.125 | 69.480 | 75.152 | 65.656 | 76.161 |
| CoCo/Insertion ↑ | **76.759** | 37.659 | 48.888 | 50.323 | 63.503 | 59.072 | 64.244 |
| ImageNet/Deletion ↓ | **17.298** | 40.784 | 30.948 | 29.903 | 21.825 | 34.191 | 19.254 |
| CoCo/Deletion ↓ | **8.318** | 28.762 | 18.422 | 17.440 | 9.950 | 15.946 | 8.426 |

Table 5: MobileNet-v2, 50 runs

|  | EAC | DeepLIFT | GradSHAP | IntGrad | KernelSHAP | FeatAbl | LIME |
|---|---|---|---|---|---|---|---|
| ImageNet/Insertion ↑ | **74.651** | 34.197 | 47.848 | 48.662 | 60.837 | 59.197 | 61.282 |
| CoCo/Insertion ↑ | **68.556** | 28.951 | 37.393 | 37.719 | 48.658 | 44.420 | 50.387 |
| ImageNet/Deletion ↓ | **6.002** | 26.381 | 14.679 | 13.382 | 7.766 | 8.866 | 7.344 |
| CoCo/Deletion ↓ | **6.684** | 21.467 | 14.237 | 14.936 | 9.308 | 11.706 | 7.106 |

Table 6: ResNet-18, 50 runs

|  | EAC | DeepLIFT | GradSHAP | IntGrad | KernelSHAP | FeatAbl | LIME |
|---|---|---|---|---|---|---|---|
| ImageNet/Insertion ↑ | **73.558** | 47.799 | 38.877 | 36.806 | 50.547 | 43.448 | 50.592 |
| CoCo/Insertion ↑ | **65.669** | 50.689 | 42.937 | 45.252 | 54.046 | 53.835 | 53.837 |
| ImageNet/Deletion ↓ | **6.596** | 8.588 | 11.273 | 11.555 | 6.638 | 8.352 | 6.776 |
| CoCo/Deletion ↓ | **5.015** | 9.097 | 11.758 | 11.483 | 7.007 | 9.325 | 6.495 |

We explore the performance of EAC on different target models. We choose three representative visual models, including ViT [48], MobileNet [49], and ResNet-18 [1], and use the same experimental setup as in the main text. We run each method for 50 times to report the average performance of each method. Overall, we observe a similar performance as shown in the main text. In particular, EAC consistently outperforms other methods on all target models.

## Backdoor Defense

Table 7: Backdoor-Defense on CIFAR-10

| ASR | Victim Model | EAC | DeepLIFT | GradSHAP | IntGrad | KernelSHAP | FeatAbl | LIME |
|---|---|---|---|---|---|---|---|---|
| BadNet [50] ↓ | 0.99 | **0.042** | 0.542 | 0.622 | 0.618 | 0.91 | 0.47 | 0.574 |
| TrojanNN [51] ↓ | 0.99 | **0.038** | 0.094 | 0.122 | 0.122 | 0.65 | 0.098 | 0.11 |

To evaluate the security impact of EAC, this section conducts backdoor removal experiments on CIFAR-10 [52]. We compare EAC and other XAI methods. Specifically, we perform two representative backdoor attacks, BadNet [50] and TrojanNN [51], on ResNet-18 as the target model. During the evaluation process, aligned with relevant works in this field [53], we remove the top three patches among every poisoned image for each XAI tool, and record the corresponding Attack Success Rate (ASR) after the removal. Overall, we randomly generate 250 poisoned images, and report their average ASR in Table 7. The evaluation results are highly encouraging; EAC has the lowest ASR under both attack settings. We interpret that this evaluation shows the high generalizability of EAC over different target models.

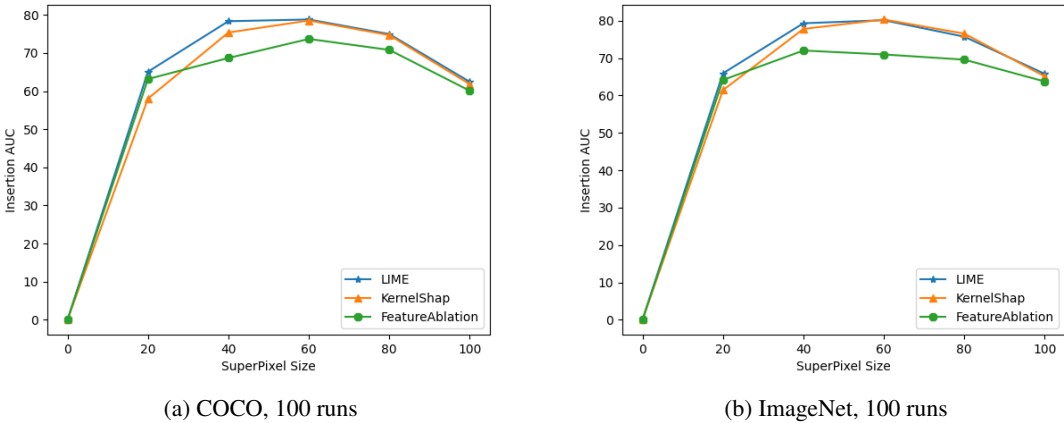

(a) COCO, 100 runs          (b) ImageNet, 100 runs

Figure 3: The effect of the superpixel size on AUC. A trade-off can be observed.

## Analysis of the Trade-off between SuperPixel Size and AUC

Overall, superpixel-based XAI tools are sensitive to the size of the superpixel. To obtain a fair comparison between EAC and de facto superpixel-based XAI tools, we carefully studied how the size of superpixel influence the performance of LIME, KernelShap, and FeatureAblation. The evaluation results using both ImageNet and COCO are shown in Fig. 3. We observed that there exists a trade-off between AUC and the superpixel size for both datasets. Empirical observation shows that a proper range of the superpixel size ranges from 40 to 80.

To unleash the full capability of superpixel-based methods, we set the superpixel size as 75 for ImageNet evaluations, and 58 for COCO evaluations, respectively, when conducting the experiments in the main paper. In contrast, EAC does *not* require such a hyperparameter tuning step, and is able to achieve superior performance over those superpixel-based methods.

