# OpenReview forum: "Explain Any Concept: Segment Anything Meets Concept-Based Explanation"
_NeurIPS.cc/2023/Conference — NeurIPS 2023 poster_

### Official Review · Reviewer_58qS · 2023-06-27

**Soundness:** 2 fair
**Presentation:** 3 good
**Contribution:** 2 fair
**Rating:** 5
**Confidence:** 4

**Summary:**

This paper proposed EAC, which lines up SAM with XAI. The technique includes there phases: (1) generates concepts with SAM; (2) trains a surrogate model to represent the target model, using the same FC layer; (3) regards results of SAM in the first phases as players and calculates Shapley values with the surrogate model. Finally, obtain the masked image based on Shapley values as the visual explanation.

**Strengths:**

1. The first one to use SAM as a concept discovery method in the concept-based XAI.
2. Good performance in the quantitative evaluation of faithfulness and user study.
3. The paper is well-organized and easy to follow.

**Weaknesses:**

1. Better to give some examples to show the trade-off. It’s hard to understand the trade-off mentioned in Sec.3.1. The highly faithful visual explanation map can show the importance of image pixels for the model to make a prediction accurately, so we use deletion/insertion to measure faithfulness. This is not conflict with human’s understanding. For example, through the explanation map from Grad-CAM or RISE, human can get where the model was looking at when did the prediction, so then human can make analysis of the model’s error mode.
And in [A, B], the user studies show that the faithfulness of the visual XAI map is consistent with human’s confidence (higher faithfulness map gets a higher rank in the user study), demonstrating that faithfulness and understandability are not conflict.

[A] Chenyang Z, Chan A B. ODAM: Gradient-based Instance-Specific Visual Explanations for Object Detection[C]//The Eleventh International Conference on Learning Representations. 2023.

[B] Petsiuk V, Jain R, Manjunatha V, et al. Black-box explanation of object detectors via saliency maps[C]//Proceedings of the IEEE/CVF Conference on Computer Vision and Pattern Recognition. 2021: 11443-11452.

2. My major concern is the significance of the work, although the paper gives good quantitative evaluation results and discussions about the potential usage of EAC.
- I’m not sure what the explanations are supposed to explain. From the samples in Figure 2, the explanation seems to be a correct segmentation of the salient object on the image, which corresponds to the classification result. The method is also tested on COCO, and I’m curious about the explanation in the situation of several same-class objects. In the fourth row, there are two zebras, and the explanation only marks out one of them; what does this mean? The model classifies the image as “zebra” because it has seen the right one, but not because of the other one?
- The explanation map is supposed to help developers to understand the predictions and analyze the model. The paper should also provide some cases of applying EAC on wrong-classified samples.
- I know that the baseline LIME also uses a surrogate model, but is that reasonable to use a surrogate model to replace the target model to be explained? How can we ensure that the same input (any coalition S of concepts) can always generate the same output between the two models? So why can use the explanation for the surrogate model to represent the explanation for the target model?

3. Line 87, the reference of GradSHAP is actually for Grad-CAM. Reference for Grad-SHAP is missing.



**Questions:**

see the "Weaknesses"

---

> ### Author Rebuttal · Authors · 2023-08-09
>
> **Q1: explain the trade-off in sec 3.1**
>
> A1: Sorry for the confusion. Our responses follow:
>
> First, the “trade-off” primarily refers to previous super-pixel based XAI methods (the mainstream), because it’s generally hard to decide an “one-size-fit-all’ superpixel size for previous works.
>
> We present the following figure, showing that superpixel size mainly decides the faithfulness and understandability. The ResNet model predicts the original image as “ballplayer,” and we use EAC with two baselines from the main paper (Superpixel-based IntGrad and cluster pixel-based GradShap) to explain this prediction.
>
> [A figure has been submitted via the "Official Comment" button to the AC]
>
> As shown in this figure, GradShap correctly identifies the player, but the cluster area is too small to be clear. Superpixel-based IntGrad is more readable than GradShap, but it is however imprecise. EAC strikes a balance by producing a well-shaped player that is easy to understand and also focuses on the ballplayer rather than irrelevant objects.
>
> Overall, a too large superpixel size can yield inaccurate yet more “complete” (and thus more readable) outputs, and vice versa for a too small superpixel size. Accordingly, the Figure 1 in our submitted Supplementary Material quantitatively reports that a too large/small superpixel size undermines the faithfulness. On the other hand, our EAC by design avoids this tradeoff problem, given that the size of the concept is intelligently extracted by SAM’s object detection.
>
> Again, thank you for your advice, and we will extend the “trade-off” paragraph to avoid confusion.
>
>
> > "And in [A, B], the user studies show that the faithfulness of the visual XAI map is consistent with human's confidence (higher faithfulness map gets a higher rank in the user study), demonstrating that faithfulness and understandability are not conflict."
>
> We wish to clarify a confusion here. Those two works empirically justify that their methods have better faithfulness & understandability than previous works. In this regard, it is aligned with our paper, because our evaluation shows that EAC achieves better faithfulness and understandability. Those two papers are however not studying the inherent tradeoff across superpixel-based methods.
>
> **Q2: Explain the “correctness” in Fig 2 of what EAC should deliver to the audiences**
>
> A2: We answer this from three aspects:
>
> Shapley Value by design measures the average expected marginal contribution of a concept after all possible combinations fairly. Namely, Shapley computes the exact importance of whether the model prefers the zebra on the left or on the right. Thus, it is technically correct to flag the right zebra as it contributes more.
>
> In fact, as we can see in Fig.2, for any baseline using Shapley Value (GradShap, KernelShap, and EAC itself), they consistently prefer the zebra on the right.
>
> Compared with other Shapley baselines using super-pixel, output by our EAC method renders a clear and well-formed zebra to the audiences, which enhances human readability a step further.
>
> **Q3: What is our rationality to use a surrogate model replacing the target model? How can we ensure that the same input (any coalition S of concepts) can always generate the same output between the two models? So why can use the explanation for the surrogate model to represent the explanation for the target model?**
>
> A3: Unlike LIME, our aim is to calculate the Shapley Value of each concept’s contribution. Computing the exact Shapley Value is computationally expensive O(2^(N)), so we take a common practice to employ a surrogate model that can estimate it [1,2,3]. To ensure that our surrogate model can produce the same output as the original model, we use Monte Carlo (MC) sampling; [2] has theoretically shown that MC makes the surrogate-provided Shapley Value closer to the true value of the target model, when more samples are made. Accordingly, we use a large number of MC samples (50000 iterations) for each experiment. Our empirical observation shows that this setting offers sufficiently good accuracy. We’ll note this in revision.
>
> [1] Graphsvx: Shapley value explanations for graph neural networks. ECML PKDD 2021.
>
> [2] Efficient Task-Specific Data Valuation for Nearest Neighbor Algorithms. VLDB 2019.
>
> [3] Shapley Computations Using Surrogate Model-Based Trees. IISA 2019.
>
>
> **Q4: Line 87, the reference of GradSHAP is actually for Grad-CAM. Reference for Grad-SHAP is missing.**
>
> A4: Sorry about this and thank you for pointing it out; we will carefully proofread the paper and fix those errors.

---

> > ### Comment · Reviewer_58qS · 2023-08-14
> >
> > Thank the author's detailed reply, which mostly solve my concerns. I would like to raise my score to 5.
> >
> > ps: I cannot see your submitted figure to AC, maybe can submit an extra PDF for the figures.

---

> > > ### Author Response · Authors · 2023-08-14
> > > **Thank you and the figure**
> > >
> > > Dear Reviewer,
> > >
> > > Thanks a lot for reading our rebuttal and raising the score! We appreciate it very much.
> > >
> > > Here is the figure:
> > >
> > > https://anonymous.4open.science/r/sam-demo-AB3D/rebuttal_demo_for_%2058qS.png
> > >
> > > This url works on our end, and kindly let us know if it does not work on your end.
> > >
> > > Sincerely,

---

### Official Review · Reviewer_bcLx · 2023-06-29

**Soundness:** 2 fair
**Presentation:** 3 good
**Contribution:** 3 good
**Rating:** 5
**Confidence:** 4

**Summary:**

This work proposes EAC to study the interpretability of models. Instead of making element-wise explanations, EAC segments an input into sub-parts, then uses Shapley value to characterize important features for a model decision. User studies were conducted to show the explainability ability of this method.

**Strengths:**

In this work, the idea that splits an input into performing sub-parts and subsequence-level explanation are reasonable. The proposed  method is also easy to understand. The user studies show the effectiveness of EAC.

**Weaknesses:**

This method is built upon LIME with certain modifications specific to approximate the target model, which compromises the technical novelty of this work. And the selection of parameters is not clear to me, which should influence the final performance. For example, how to choose an optimal surrogate model suitable for datasets across different domains. Besides, the comparison methods in Tables 1 & 2 are vanilla so that they not quite convincing.  Maybe more stronger methods need to be considered. In addition, the baseline methods, i.e., LIME and KernelSHAP, used in Tables 1 & 2 are proposed in 2016 and 2017 respectively. More recent methods should be considered as baselines, for example, “On locality of local explanation models, NeurIPS 2021”, “Craft: Concept recursive activation factorization for explainability, CVPR 2023”, “RKHS-SHAP: Shapley values for kernel methods, NeurIPS 2022”. Please address the weakness parts in the rebuttal.


**Questions:**

This paper is well presented with an interesting idea based on Shapley value. Since it’s mainly based on Shapley value technique, this may limit the originality contribution.

**Limitations:**

The authors discussed broader societal impacts.

---

> ### Author Rebuttal · Authors · 2023-08-09
>
> **Q1: EAC built upon LIME, compromise technical novelty**
>
> A1:  Thanks for the comment. Indeed, we found that the binary feature expression by LIME is very inspiring, such that we adopted those expressions in our pipeline. Nevertheless, there are two main major differences between our work EAC and LIME: 1) Concept-wise SAM+Shapley and 2) Per-Input Equivalence (PIE).
>
> 1）Unlike LIME, we choose to use Shapley Value as our importance score measurement. This is because Shapley Value accounts for the average expected marginal contribution of a concept after all possible combinations fairly. In contrast, LIME only uses Super-Pixel for concept discovery, which is often less human-friendly and understandable than SAM extractor, as shown in Fig.2.
>
> 2）To reduce the heavy computational burden of Shapley Value, we propose the scheme of Per-Input Equivalence (PIE). Moreover, while LIME only uses a set of linear weights as the surrogate model, ours not only mimics the feature extractor but also retains its original fully-connected layer.
>
> Also, besides the above *technical novelty* comparison, we wish to clarify that our paper has *conceptual-level novelty* (i.e., we for the first time advocate using SAM as a concept discovery method to facilitate concept-based XAI and achieve high faithfulness and understandability) and also *highly encouraging empirical results*. We’ll better clarify our contributions in the revision.
>
> **Q2: unclear selection of hyperparam for EAC?**
>
> A2: Thanks for the question. We clarify that EAC does not require much “hyper-param.” The only hyper-param involved is when fitting the PIE Scheme, i.e. a simple linear neural network learning scheme and the Monte Carlo (MC) sampling: we set `lr=0.008` and the number of MC sampling as `50000` throughout all experiments. We’ll clarify this in the revision, and we also explain why using a MC sampling threshold of `50000` when answering the **Q2** of reviewer 6va6 above.
>
> **Q3: more and stronger baselines**
>
> A3:
> | SOTA  | Imagenet adding (higher the better) | Imagenet removing (lower the better) | COCO adding(higher the better) | Coco removing(lower the better) |
> |-------|-------------------------------------|--------------------------------------|--------------------------------|---------------------------------|
> | EAC   | 83.40                               | 23.799                               | 83.404                         | 16.640                          |
> | CRAFT | 60.40                               | 54.66                                | 51.49                          | 44.93                           |
>
>
> Following your suggestion, we launch more experiments to compare with baselines during the rebuttal phase. In particular, we report the performance of Craft (CVPR’23) compared with EAC under the same experiment setting in table 1. For Craft, we report its best performance after carefully tuning its hyper-param: patch_size=32, num_super-pixel=75/58 imagenet/coco. EAC outperforms CRAFT by 20 AUC percentage points in almost all cases.
>
> On the other hand, RKHS-SHAP and Local Explanation are originally designed for numerical data with two dimensions (rows and columns). To adapt them for image tensors with four or three dimensions would take too much time and effort during the rebuttal timing window. We attempted to use the Local Explanation Models pipeline by choosing only the last fully-connected layer of the nn model, which produces a two-dimensional feature vector of size `n x 2048`, where `n` is the number of superpixel concepts. However, this did not work because the Local Explanation Models pipeline needs `n` to be larger than 2048, which is not feasible.

---

> > ### Comment · Reviewer_bcLx · 2023-08-17
> >
> > Thank you very much for this extensive and informative rebuttal. I would like to raise my score to 5.

---

### Official Review · Reviewer_6va6 · 2023-07-05

**Soundness:** 3 good
**Presentation:** 3 good
**Contribution:** 3 good
**Rating:** 5
**Confidence:** 4

**Summary:**

This article proposes EAC, which aims to use the Segment Anything Model to generate some prior concepts. By constructing a surrogate model, the concept combination area most relevant to the decision category is calculated by Shapley Value. The author evaluates the proposed model from three perspectives: faithfulness, understandability and effectiveness. Experimental results demonstrate the superiority of the proposed method.

**Strengths:**

- SAM-based prior methods will provide more accurate localization on conventional data than superpixel-based methods.
- This approach can explain both CNN and ViT models (in supplementary material).
- The authors provided the code to ensure reproducibility.

**Weaknesses:**

- The authors compare DeepLIFT, GradSHAP, and IntGrad methods, which use superpixel-based methods. Can the authors replace the superpixel method with the input result of Segment Anything to make the method comparison fairer?
- Why is it necessary to train a surrogate model? Is this to improve inference efficiency?
- Since this paper only uses SAM to generate prior knowledge, and then uses Shapley Value to estimate the importance score of prior concepts. I would like to see a concrete comparison with other Shapley Value based methods such as GradShap+SAM or FastShap+SAM.

**Questions:**

- Are the deletion and insertion indicators used by the author deleted according to the score of shapley value?
- Since the results generated by SAM have no labels, can this segmentation result be called a concept?

**Limitations:**

The authors have discussed potential social implications.

---

> ### Author Rebuttal · Authors · 2023-08-09
>
> **Q1: compare our results with  GradShap+SAM or FastShap+SAM.**
>
> A1:
>
> Thank you for this insightful question. We indeed explored this direction before, and our preliminary observation shows that this is unpromising; please see our response to **Q2** below on the conceptual-level clarification of “training a surrogate model.”
>
> Following your suggestion, below we launch more experiments to compare with baselines. Due to the large baselines for evaluation and the short timing window during rebuttal, we focus on evaluating the "Adding" operation: adding and removing operations share conceptually similar difficulty, as reflected in our results in the paper (tools behave consistently under "Adding" and "Removing" operations; as shown in Table 1, tools with better score in adding also preserve approximately the same performance ranking in deleting, such as EAC, LIME, DeepLift).
>
> | SOTA         | Imagenet adding | COCO adding |
> |--------------|-----------------|-------------|
> | KernelSHAP+sam  | 81.76           | 75.65       |
> | DeepLIFT+sam | 52.82           | 49.27       |
> | LIME+sam     | 79.85           | 77.50       |
> | EAC (Ours)     | 83.40           | 83.404      |
> | FeatAbl+sam  | 72.74           | 71.24       |
> | GradSHAP+sam | 44.47           | 41.37       |
>
> From the table above, we interpret that our EAC method consistently achieves the best result and outperforms others. In the COCO dataset, it is even six percentages higher than that of the second best (lime+sam). This empirically justifies the advantage of employing our customized lightweight surrogate model.
>
> We’ll add the above results (with further evaluations on “Removing”) in revision.
>
> **Q2: Why is it necessary to train a surrogate model? Is this to improve inference efficiency?**
>
> A2: First, using a surrogate model appears to be a common practice in this field of research [1,2,3]; for instance, the seminal work, LIME, also uses surrogate models to approximate a neural network’s decision and ease XAI.
>
> Nevertheless, Unlike LIME, the surrogate model in our work mainly serves to efficiently compute the Shapley value of each concept’s contribution (thus to answer your question: yes, it’s mainly for “improving inference efficiency”).
>
> Overall, computing the exact Shapley Value is computationally expensive O(2^(N)), so we take a common practice to employ a surrogate model that can estimate it [1,2,3]. To ensure that our surrogate model can produce the same output as the original model, we use Monte Carlo (MC) sampling; [2] has theoretically shown that MC makes the surrogate-provided Shapley Value closer to the true value of the target model, when more samples are made. Accordingly, we use a large number of MC samples (50000 iterations) for each experiment. Our empirical observation shows that this setting offers sufficiently good accuracy. We’ll note this in revision.
>
> [1] GraphSVX: Shapley Value Explanations for Graph Neural Network. ECML PKDD 2021.
>
> [2] Efficient Task-Specific Data Valuation for Nearest Neighbor Algorithms. VLDB 2019.
>
> [3] Shapley Computations Using Surrogate Model-Based Trees. IISA 2019.

---

> > ### Comment · Reviewer_6va6 · 2023-08-16
> > **Another Question**
> >
> > Thanks, how were the visualized regions in Figure 2 selected? Is it based on a threshold? Or the most important segmented region to choose?

---

> > > ### Author Response · Authors · 2023-08-16
> > >
> > > Dear reviewer,
> > >
> > > We appreciate your response to our rebuttal. All the SOTA methods in Fig2 produce an importance score for each patch in an image, using either Shapley, Gradient, or other approaches. We then choose the concept patch with the highest score for each SOTA method. Sorry for the confusion, We’ll clarify this in revision.

---

### Official Review · Reviewer_xxGh · 2023-07-07

**Soundness:** 3 good
**Presentation:** 3 good
**Contribution:** 3 good
**Rating:** 6
**Confidence:** 3

**Summary:**

The paper introduces Explain Any Concept (EAC), a concept-based explanation approach that enhances the interpretability of deep neural networks in computer vision. EAC utilizes the Segment Any Model (SAM) as a concept discovery technique, and the authors propose a lightweight Per-Input Equivalence (PIE) scheme that employs a proxy model to approximate the target model. This scheme reduces computational costs while maintaining reasonable fidelity, effectively addressing the challenges related to fidelity, comprehensibility, and efficiency in Explainable Artificial Intelligence (XAI). The effectiveness of EAC is demonstrated through experiments conducted on the ImageNet and COCO datasets, where it outperforms pixel-level and superpixel-based methods in terms of accuracy and interpretability. The authors emphasize the potential applications of EAC in various domains, including medical image analysis, autonomous driving, and robotics.

**Strengths:**

The EAC is a novel concept-based interpretable method that integrates the recently released SAM. It leverages the zero-shot/few-shot capabilities of the segment anything model, addressing the limitations of pixel-based methods and the need for human annotation in concept-based approaches.
The proposed lightweight per-input equivalent (PIE) scheme improves the efficiency of the explanation process while maintaining high faithfulness and understandability.
The paper conducts extensive experiments and evaluations on popular datasets such as ImageNet and COCO to demonstrate the effectiveness of EAC. Comparisons with traditional pixel-level and superpixel-based XAI methods showcase the superior performance of EAC.

**Weaknesses:**

All datasets used in the paper are natural images in the general domain. However, it is important to include some knowledge-specific domains, such as medical images, for evaluation purposes.
The performance of the proposed methods can be affected by the SAM segmentation performance. It has already been found that SAM struggles to segment medical images or tiny-scale objects.
Some failure cases should be included to provide a comprehensive overview of the model's faithfulness.

**Questions:**

May need proofreading:
All references in Supplementary Material are shown incorrectly.

**Limitations:**

Lack of real-world deployment evaluations: While this paper discusses the potential impact and applications of EAC in various domains, it does not provide real-world deployment evaluations or case studies to demonstrate the practical effectiveness of EAC in real-world scenarios.

---

> ### Author Rebuttal · Authors · 2023-08-09
>
> **Q1: why not include some knowledge-specific domain dataset to eval?**
>
> A1: Thank you for your comments, and we agree with you. Yes, our observation and exploration also show that the Meta SAM model was only trained on the general image domain, and it may struggle to segment images in knowledge-specific domains.
>
> And from a more general perspective, a common challenge for neural models is their poor generalization in domains that require specific knowledge. Besides SAM, even the latest and most advanced Large Language Models (LLMs) like ChatGPT/LLAMA suffer from this issue as well [1,2,3].
>
> Having that said, with the recent development of knowledge-specific area SAM such as medical image [4], remote sensing [5], and UVA [6], we believe EAC has the potential to improve DNN decisions on other targeted areas. We leave it as one future work, as discussed in Section 6. Since this is the first attempt to bring humanly understandable and computationally feasible concept extractors to the field XAI, we believe our work can shed light on their explainability.
>
> In revision, we would like to follow your suggestions to clarify this point (SAM’s limitation) in the paper; this shall better provide a fair and comprehensive overview of the model’s faithfulness without potential overclaims.
>
>
> [1]: Harnessing the Power of LLMs in Practice: A Survey on ChatGPT and Beyond.
>
> [2]: Is ChatGPT a good translator? A preliminary study.
>
> [3]: The ConceptARC Benchmark: Evaluating Understanding and Generalization in the ARC Domain.
>
> [4]; SAM on Medical Images: A Comprehensive Study on Three Prompt Modes.
>
> [5]: The Segment Anything Model (SAM) for Remote Sensing Applications: From Zero to One Shot.
>
> [6]: SAM-DA: UAV Tracks Anything at Night with SAM-Powered Domain Adaptation.

---

> > ### Comment · Reviewer_xxGh · 2023-08-20
> >
> > Thank you for your feedback. I'm happy with the clarification, and I will keep the weak acceptance.

---

### Decision · Program_Chairs · 2023-09-21

**Decision:**

Accept (poster)

**Comment:**

This paper introduces Explain Any Concept (EAC), a method that leverages Segment Any Model (SAM) to enhance Explainable AI (XAI) for deep neural networks. SAM is used to provide instance segmentation for any concept and is incorporated into the Shapley method. A per-input equivalent (PIE) scheme is introduced to approximate SAM computation with a lightweight surrogate model. Experiments were conducted on ImageNet and COCO.

The paper initially received mixed reviews, with two reviewers recommending acceptance and three reviewers leaning towards rejection. The main concerns raised by reviewers related to the application domain of EAC, its positioning with respect to LIME, the discussion on faithfulness vs. understandability, and the need for clarification regarding experiments (such as stronger baselines, using SAM with other XAI baselines, and setting hyper-parameters). The rebuttal effectively addressed these concerns, and the three most negative reviewers increased their ratings to 5. As a result of the authors' feedback, a consensus among reviewers was reached for a (weak) paper acceptance.

The AC has thoroughly reviewed the submission and the discussions. Although the proposed method may appear incremental, the AC finds the combination of SAM and Shapley for XAI timely and well-conducted and described. The AC is convinced that this solution may be of great interest to the community and inspire follow-up works in explaining any concept. The AC thus recommends paper acceptance and encourages the authors to include the rebuttal in the final version of the paper.